# Major Adverse Events in Patients with Peripheral Artery Disease after Endovascular Revascularization: A Retrospective Study

**DOI:** 10.3390/jcm11092547

**Published:** 2022-05-01

**Authors:** Mihui Kim, Yong Sook Yang, Young-Guk Ko, Mona Choi

**Affiliations:** 1College of Nursing, Brain Korea 21 FOUR Project, Yonsei University, Seoul 03722, Korea; mihuikim@yuhs.ac (M.K.); ysyang100061@gmail.com (Y.S.Y.); 2Division of Cardiology, Severance Cardiovascular Hospital, Yonsei University College of Medicine, Seoul 03722, Korea; ygko@yuhs.ac; 3Mo-Im Kim Nursing Research Institute, College of Nursing, Yonsei University, Seoul 03722, Korea

**Keywords:** diabetes mellitus, exercise, major adverse cardiovascular event, major adverse limb event, peripheral artery disease

## Abstract

Objective: For peripheral artery disease (PAD) patients, after endovascular revascularization, it is crucial to manage associated factors that can affect the risk of major adverse events. We aimed to investigate the associated factors of major adverse events in these patients. Materials and Methods: We conducted a retrospective longitudinal analysis using the electronic medical records from a tertiary hospital in Korea and included the data of 1263 patients. Eligible patients were categorized into four groups based on diabetes mellitus (DM) and regular exercise. The major adverse events included major adverse limb events and major adverse cardiovascular events. Major adverse events-free survival was assessed using the Kaplan–Meier method, and associated factors of major adverse events were analyzed using Cox proportional hazards analyses. Results: Kaplan–Meier survival curves showed that patients with DM and non-regular exercise had a shorter major adverse events-free survival. The Cox regression analysis showed that for patients with critical limb ischemia or chronic kidney disease, the risk of major adverse events increased, while group variables were not significant. Conclusion: Target management of patients with DM, critical limb ischemia, and chronic kidney disease is essential to reduce major adverse events after endovascular revascularization in patients with PAD.

## 1. Introduction

Peripheral artery disease (PAD) is defined as an atherosclerotic arterial disease that reduces blood flow to the affected limb by stenosis or occlusion of the arteries [1,2,3]. The prevalence of PAD was estimated to be 4.6% in the general population in South Korea [4] and affected 236.62 million patients worldwide in 2015, rising steeply with aging [5].

The manifestation of PAD may be asymptomatic, or there may be symptoms such as intermittent claudication, atypical leg pain, and critical limb ischemia [2,6,7]. However, regardless of symptoms, patients with PAD may experience an impaired walking ability and functional status and poor quality of life, which further increase the risk of cardiovascular ischemic events, limb-related events, and mortality [2,7,8]. Endovascular revascularization is the preferred procedure for relieving the symptoms of PAD according to lesion characteristics [2,9]. 

Major risk factors for PAD are similar to cardiovascular risk factors, such as diabetes mellitus (DM), smoking, dyslipidemia, and hypertension; among these factors, DM and cigarette smoking have the strongest risks [7,10]. DM, a known metabolic disease, promotes local inflammatory reactions on vascular walls and reduces peripheral blood flow, which may result in a disorder of endothelial cell function (nitric oxide mechanism) and vascular control [11]. DM significantly increases adverse clinical outcomes [5,12,13] and induces restenosis after endovascular revascularization [14]. The incidence of DM has quadrupled worldwide over the past 30 years [15], which indicates that the burden of PAD due to DM is likely to increase. Meanwhile, the cigarette-smoking population steadily declined worldwide between 2000 and 2015 [16]. 

Several studies showed that regular exercise improved the walking ability, functional status, and overall quality of life of patients with PAD [2,17,18]. One study found that regular exercise improved metabolic function, the levels of C-reactive protein, and the ankle–brachial index [18]. Other studies showed that regular exercise positively affected clinical outcomes by increasing oxygen uptake in the lower extremities [17,19]. However, the impact of regular exercise on major adverse events after revascularization of PAD patients remains unknown. 

Major adverse events are defined as major adverse limb events (MALE) and major adverse cardiovascular events (MACE), which are indicators of long-term outcomes in PAD [20,21]. Therefore, this study investigated the associated factors of MALE and MACE in patients with PAD after endovascular revascularization using electronic medical records (EMRs).

## 2. Materials and Methods

### 2.1. Data Source and Study Patients

We conducted a retrospective longitudinal analysis using EMR data from a tertiary hospital in Seoul, Korea, between January 2009 and December 2018. The EMRs contain inpatient and outpatient clinical records of PAD patients. In this study, we excluded patients diagnosed with PAD before 2009 and considered 3852 patients who were first diagnosed with PAD between 2009 and 2018, of which 1288 underwent endovascular revascularization. Of these, we excluded 25 patients because they had undergone either bypass surgery or major amputation before 2009. The remaining 1263 eligible patients were then categorized into four groups according to the following criteria: whether they were diagnosed with DM and whether they exercised regularly (Group A: non-DM/regular exercise, Group B: non-DM/non-regular exercise, Group C: DM/regular exercise, and Group D: DM/non-regular exercise). The flow of the cohort derivation is shown in Figure 1. 

### 2.2. Baseline Variables

Baseline demographic and clinical data, including comorbidity, disease severity, initial nursing assessment, physician’s progress notes, and procedure record, were extracted from the EMRs. We defined the date of the first endovascular revascularization as the index date for each patient who was hospitalized to undergo endovascular revascularization. The initial nursing assessment included the patient’s past medical history, medications, activities, and psychosocial status on the day of admission [22]. Baseline regular exercise data were extracted from the initial nursing assessment record gathered from a self-reported question regarding whether the patient exercised regularly. Clinical manifestations (asymptomatic, claudication, critical limb ischemia, and abnormal skin color) were recorded as either Rutherford stage or narrative in the physicians’ progress notes, procedure records, and initial nursing assessment. Abnormal skin color recorded in physicians’ progress notes means a color change in toes without symptoms.

We extracted the past medical history (DM, hypertension, dyslipidemia, chronic kidney disease, coronary artery disease, congestive heart failure, and ischemic stroke) from the physicians’ progress notes using the International Classification of Diseases, 10th Revision, Clinical Modification (ICD-10-CM) codes. Additionally, DM was defined by ICD-10-CM codes and by a 6.5% or higher level of glycosylated hemoglobin (HbA1c). 

### 2.3. Major Adverse Events

The major adverse events included MALE and MACE. MALE was defined as a composite of major amputation, bypass surgery, repeat endovascular revascularization, and in-hospital death. The procedure records were used to confirm repeat endovascular revascularization, and death records were used to determine in-hospital death. MACE was defined as a composite of myocardial infarction, ischemic stroke, and in-hospital death [2]. 

In the hospital used for EMR analysis, the surgical/procedure code lists and disease diagnosis were recorded using International Classification of Diseases, 9th Revision, Clinical Modification (ICD-9-CM) and ICD-10-CM, respectively. Therefore, ICD-9-CM was used to identify major amputation (8414, 8415, 8417) and bypass surgery (3925, 3929) from the operation record. The ICD-10-CM codes were used to identify myocardial infarction (I21, I22, I23, I241, and I252) and ischemic stroke (I63) from the physicians’ progress notes. Time to events was measured in years from the index date to the first major adverse event’s incidence.

### 2.4. Statistical Analysis

Descriptive statistics were used for the clinical characteristics of the groups. The chi-square test or Fisher’s exact test and analysis of variance (ANOVA) were used for group differences. The Kaplan–Meier survival analysis was performed for MALE- and MACE-free survival curves, and log-rank tests were carried out for differences between the groups. A Cox proportional hazards regression model determined associated factors of MALE and MACE incidence. The proportional hazards assumption was tested using the Schoenfeld residual analysis. The hazard ratio (HR) was provided with a 95% confidence interval (CI). For all tests, a *p*-value < 0.05 was considered statistically significant. Data analysis was performed using IBM SPSS Statistics for Window, Version 25.0 (Armonk, NY, USA) and RStudio (version 1.3.1056, RStudio, PBC, Boston, MA, USA).

## 3. Results

### 3.1. Characteristics of the Study Patients

A total of 1263 patients were classified into four groups based on DM and regular exercise, with their baseline characteristics presented in Table 1. More than half of the total patients had DM; 36.1% did not exercise regularly, and 18.1% exercised regularly. Approximately 46% of the patients had no DM, of which 29.0% did not exercise regularly, and 16.8% exercise regularly. The mean age of patients was 67.3 ± 11.7 years, and 80.8% were men. A total of 1207 patients experienced symptoms of clinical manifestation, such as intermittent claudication (59.6%), critical limb ischemia (32.2%), and abnormal skin color (4.5%). The numbers of patients diagnosed with hypertension, DM, and dyslipidemia were 864 (68.4%), 685 (54.2%), and 570 (45.1%), respectively.

There was a significant group difference in multilevel disease (*p* = 0.002). The aortoiliac lesions were more frequent in Group A, while infrapopliteal lesions were more frequent in Group D. The femoropopliteal lesions did not differ between the groups (*p* = 0.341). Balloon angiography was performed in approximately 97%, while stent insertion was performed in 56.6%.

### 3.2. Major Adverse Events

The total number of patients with MALE and MACE during the study period is summarized in Table 2. The total number of patients with MALE was 465 (36.8%), and those with MACE numbered 158 (12.5%). MALE occurred most frequently in Group D (DM/non-regular exercise, 42.8%), with significant group differences (χ^2^ = 29.47, *p* < 0.001). MACE also occurred most frequently in Group D (50.0%), with significant group differences (χ^2^ = 27.42, *p* < 0.001). Major amputation, endovascular revascularization, death, and myocardial infarction occurred most frequently in Group D, with significant differences between groups.

The Kaplan–Meier survival curves for MALE- and MACE-free survival showed significant group differences (log-rank test, *p* < 0.001). Group D showed the poorest prognosis in the MALE- and MACE-free survival (Figure 2).

### 3.3. Hazard Ratios of MALE and MACE

Before analyzing the Cox regression model, we tested the proportional hazards assumption using the Schoenfeld residuals test. The results show no violation of the proportional hazards assumption (MALE, *p* = 0.216; MACE, *p* = 0.089); therefore, we estimated the HR with 95% CI using the Cox regression model. We performed a univariate analysis for factors that affected the major adverse events (Table 3). For MALE, patients with critical limb ischemia had a 2.16 times higher incidence (95% CI 1.79–2.61) than those without. The MACE incidence was 2.07 times higher in Group C (DM/regular exercise; 95% CI 1.18–3.64) and 2.73 times higher in Group D (DM/non-regular exercise; 95% CI 1.63–4.56) compared to Group A (non-DM/regular exercise). Characteristics of groups, higher age, sex (female), critical limb ischemia, chronic kidney disease, congestive heart failure, and longer DM duration were the factors associated with an increased risk of MALE and MACE. In addition, hypertension (HR = 1.83, 95% CI 1.25–2.69) and coronary artery disease (HR = 1.63, 95% CI 1.19–2.23) also increased the risk of MACE.

The Cox regression analysis for multivariate analysis showed that patients with critical limb ischemia, chronic kidney disease, and congestive heart failure had 1.85-, 1.40-, and 1.36-times higher risks of MALE, respectively (Table 4). The incidence of MACE was 2.77 times higher in patients with chronic kidney disease. The major adverse events incidence was not statistically significant between the four groups.

## 4. Discussion

This study investigated the associated factors of MALE and MACE after endovascular revascularization in PAD patients based on the grouping by DM and regular exercise. In the present study, group differences were not statistically significant in MALE and MACE incidence after endovascular revascularization. Target lesion status (location, length of lesions, and degree of calcification) or clinical ischemia before the endovascular revascularization affects the vessel patency [2] associated with exercise maintenance and clinical prognosis. Thus, the CLI and CKD associated with the vascular condition before the procedure may be at high risk for the incidence of MALE and MACE. The risk factors for major adverse events found in this study were critical limb ischemia, chronic kidney disease, and congestive heart failure for MALE, and higher age, critical limb ischemia, and chronic kidney disease for MACE. These results are similar to those of previous studies identifying the risk factors for adverse clinical outcomes of PAD [2,6,23,24]. In particular, chronic kidney disease, a common independent predictor of major adverse events, is consistent with poorer outcomes in patients with PAD [25,26].

The multivariate analysis showed that the DM and non-regular exercise group had a higher risk of MALE and MACE than the non-DM and regular exercise group, but it was not statistically significant. There were significant group differences in MALE- and MACE-free survival curves, and the DM and non-regular exercise group had a shorter major adverse event-free survival than others. In the univariate analysis, the DM and non-regular exercise group had a higher risk of major adverse events with statistical significance. Direct comparisons with previous studies are difficult because there is no study on the association between DM, regular exercise, and major adverse events in patients with PAD; however, a cohort study that examined PAD patients after 30 days post-revascularization showed the increased likelihood of a major adverse event for an increased frailty score [27] affected by physical activity and exercise [28]. Although our study did not show statistical significance, regular exercise may be associated with decreased major adverse events. It is necessary to investigate the relationship between major adverse events and exercise habits in future studies.

In this study, 34 patients (5.0%) in the DM groups and 11 patients (1.9%) in the non-DM groups underwent major amputation. In addition, non-DM groups also had a lower incidence of MALE and MACE than DM groups (*p* < *0*.001). In the previous studies, regular exercise was shown to decrease overall mobility loss and lower limb amputation and maintain the functional status and walking performance of patients [2,17,19]. Therefore, healthcare providers need to ensure early detection and adequate management of people at risk to lower the incidence of major adverse events by encouraging regular exercise.

Our study showed that 36.8% of the patients experienced MALE, and 12.5% experienced MACE during the follow-up. In a previous study using healthcare claims data, patients with PAD experienced MALE (22.9%) and MACE (11.3%) during a mean of a 1.8-year follow-up period [29]. A five-year follow-up study with PAD patients using EMR data reported that 38.2% experienced MALE, and 17.0% experienced MACE [6]. The incidence of MALE was almost three times that of MACE in this study, which was similar to other studies, despite slight differences in the composition of diseases in MALE and MACE and the study samples.

Our study had several limitations. First, we extracted regular exercise data from self-reported data recorded in the EMRs. Self-reported regular exercise data can be used as an indicator of patients’ exercise habits, but more objective indicators of frequency, intensity, and duration of exercise were not reflected. However, EMRs contain a vast number of real-world data used in many studies investigating the clinical course of diseases and predicting adverse clinical outcomes [6,30]. Therefore, this study is meaningful because it used EMR data to investigate long-term clinical outcomes in PAD patients. Second, as the study data were collected from a tertiary hospital’s EMRs, the generalizability may be limited; however, this tertiary hospital adopted a full EMR system in which rich clinical documents are stored electronically [31]. Finally, as this was a retrospective longitudinal study, we could not control other potential confounding variables during the follow-up period. In addition, in this study, due to the limitations of the relevant information stored in EMRs, clinical manifestations, not Rutherford or Fontaine classifications, which classify symptoms in PAD patients, were reported.

## 5. Conclusions

As far as we know, this is the first study to investigate the associated factors of MALE and MACE after endovascular revascularization in patients with PAD focusing on DM and regular exercise. Critical limb ischemia and chronic kidney disease increase the risks of MALE and MACE in patients with PAD. Further study is needed to explore this finding and determine the long-term effects of the trajectory change in exercise habits and DM management. Additionally, in clinical practice, target management of PAD patients with DM, critical limb ischemia, and chronic kidney disease is essential to reduce the risk of major adverse events after endovascular revascularization. 

## Figures and Tables

**Figure 1 jcm-11-02547-f001:**
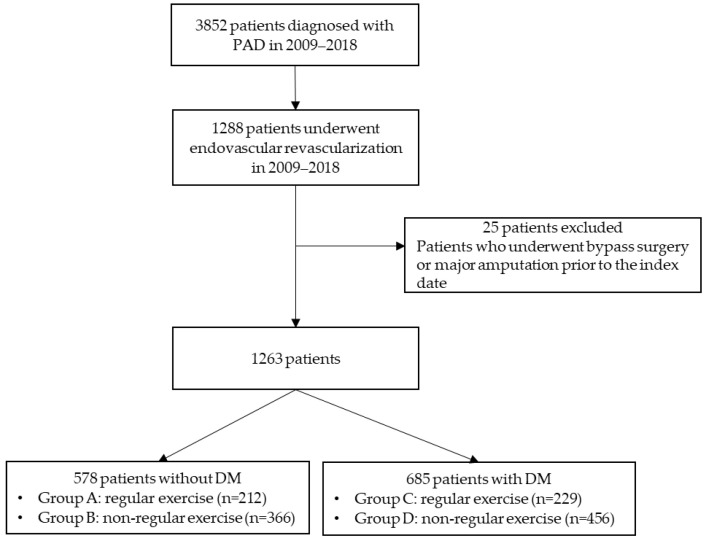
Flow chart of this study. DM, diabetes mellitus; PAD, peripheral artery disease.

**Figure 2 jcm-11-02547-f002:**
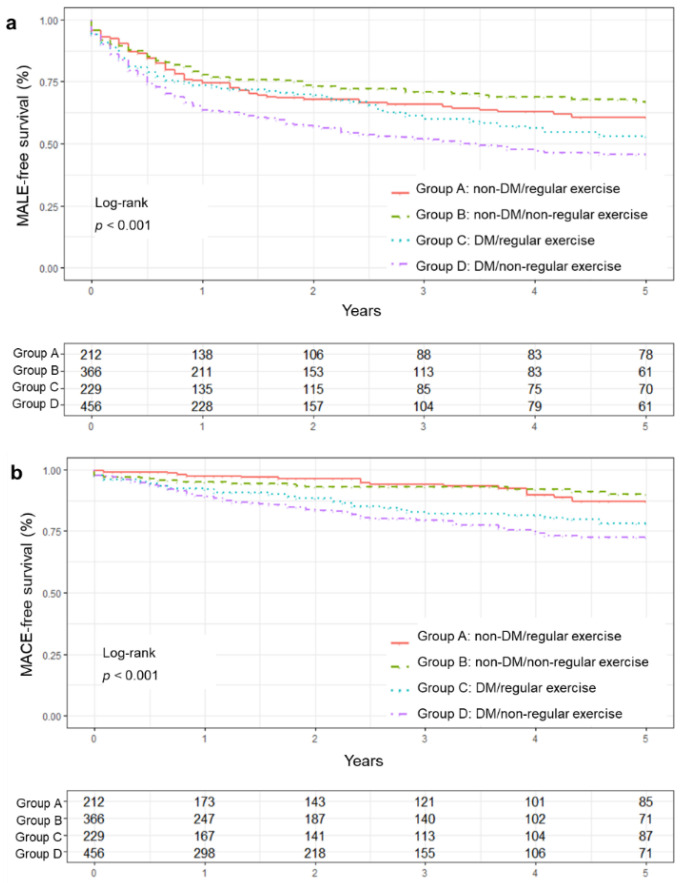
Major adverse events-free survival curve between groups. (**a**) MALE-free survival curve, (**b**) MACE-free survival curve. DM, diabetes mellitus; MACE, major adverse cardiovascular events; MALE, major adverse limb events.

**Table 1 jcm-11-02547-t001:** Baseline characteristics between groups (*n* = 1263).

Variables	Total	Non-DM (*n* = 578)		DM (*n* = 685)		χ^2^ or F	*p*
		Regular Exercise	Non-Regular Exercise	Regular Exercise	Non-Regular Exercise		
		Group A	Group B	Group C	Group D		
		(*n* = 212)	(*n* = 366)	(*n* = 229)	(*n* = 456)		
Age, years	67.3 ± 11.7	65.3 ± 12.7	66.0 ± 14.1	67.4 ± 9.1	69.1 ± 9.7		
<65	447 (35.4)	92 (20.6)	135 (30.2)	84 (18.8)	136 (30.4)	7.82	<0.001
≥65	816 (64.6)	120 (14.7)	231 (28.3)	145 (17.8)	320 (39.2)		
Sex							
Male	1020 (80.8)	178 (17.5)	300 (29.4)	186 (18.2)	356 (34.9)	3.90	0.273
Female	243 (19.2)	34 (14.0)	66 (27.2)	43 (17.7)	100 (41.1)		
BMI, kg/m^2^	23.3 ± 3.3	23.2 ± 3.3	23.4 ± 3.5	23.4 ± 2.8	23.3 ± 3.4		
Underweight (<18.5)	77 (6.2)	12 (15.6)	21 (27.3)	10 (13.0)	34 (44.1)	0.34	0.756
Normal (18.5–24.9)	807 (65.0)	139 (17.2)	227 (28.1)	152 (18.9)	289 (35.8)		
Overweight (≥25.0)	357 (28.8)	57 (16.0)	106 (29.7)	62 (17.3)	132 (37.0)		
Smoking							
Current smoker	344 (27.3)	78 (22.7)	124 (36.0)	45 (13.1)	97 (28.2)	33.05	<0.001
Former smoker	439 (34.8)	68 (15.5)	117 (26.7)	87 (19.8)	167 (38.0)	3.61	0.308
Clinical manifestation							
Asymptomatic	47 (3.7)	16 (34.1)	9 (19.1)	15 (31.9)	7 (14.9)	21.47	<0.001
Intermittent claudication	747 (59.6)	159 (21.3)	259 (34.7)	114 (15.2)	215 (28.8)	79.75	<0.001
Critical limb ischemia	404 (32.2)	22 (5.4)	78 (19.3)	89 (22.1)	215 (53.2)	117.48	<0.001
Abnormal skin color	56 (4.5)	14 (25.0)	16 (28.6)	9 (16.1)	17 (30.3)	3.02	0.391
Aspirin	1144 (90.6)	181 (15.8)	344 (30.1)	186 (16.3)	433 (37.8)	45.44	<0.001
Clopidogrel	1049 (83.1)	174 (16.6)	291 (27.7)	199 (19.0)	385 (36.7)	6.43	0.092
Statin	1179 (93.3)	194 (16.4)	338 (28.7)	213 (18.1)	434 (36.8)	4.24	0.237
Comorbidity							
Hypertension	864 (68.4)	113 (13.1)	217 (25.1)	164 (19.0)	370 (42.8)	71.76	<0.001
Dyslipidemia	570 (45.1)	68 (11.9)	168 (29.5)	86 (15.1)	248 (43.5)	35.76	<0.001
Chronic kidney disease	279 (22.1)	20 (7.2)	39 (14.0)	58 (20.8)	162 (58.0)	96.76	<0.001
Coronary artery disease	482 (38.2)	74 (15.3)	102 (21.2)	109 (22.6)	197 (40.9)	30.93	<0.001
Congestive heart failure	109 (8.6)	4 (3.7)	26 (23.8)	17 (15.6)	62 (56.9)	27.99	<0.001
Ischemic stroke	111 (8.8)	12 (10.8)	27 (24.3)	26 (23.4)	46 (41.5)	6.34	<0.001
Run-off vessels							
Multilevel disease ^†^	428 (33.9)	60 (28.3)	143 (39.1)	59 (25.8)	166 (36.4)	15.38	0.002
Aortoiliac	485 (38.4)	108 (50.9)	182 (49.7)	70 (30.6)	125 (27.4)	63.16	<0.001
Femoropopliteal	758 (60.0)	122 (57.5)	224 (61.2)	128 (55.9)	284 (62.3)	3.35	0.341
Infrapopliteal	486 (38.5)	48 (22.6)	120 (32.8)	94 (41.0)	224 (49.1)	49.93	<0.001
Endovascular technique							
Balloon angiography	1232 (97.5)	202 (95.3)	354 (96.7)	226 (98.7)	450 (98.7)	9.29	0.026
Stent insertion	715 (56.6)	135 (63.7)	235 (64.2)	116 (50.7)	229 (50.2)	23.80	<0.001
Duration of DM, years		-	-	2.6 ± 3.2	3.4 ± 4.4	-	0.005

Notes: Data are expressed as the mean ± SD or *n* (%), ^†^ two or more different lesions, BMI, body mass index; DM, diabetes mellitus; SD, standard deviation.

**Table 2 jcm-11-02547-t002:** Major adverse events between groups (*n* = 1263).

Major Adverse Events	Variables	Total	Group A	Group B	Group C	Group D	χ^2^	*p*
MALE		465 (36.8)	77 (16.6)	95 (20.4)	94 (20.2)	199 (42.8)	29.47	<0.001
	Major amputation	45 (3.6)	4 (8.9)	7 (15.6)	11 (24.4)	23 (51.1)	8.571	0.034
	Repeated endovascular revascularization	352 (27.9)	56 (15.9)	70 (19.9)	74 (21.1)	152 (43.1)	23.17	<0.001
	Surgical bypass	51 (4.0)	13 (25.5)	17 (33.3)	11 (21.6)	10 (19.6)	7.10	0.067
	Death	113 (8.9)	15 (13.3)	14 (12.4)	25 (22.1)	59 (52.2)	22.71	<0.001
MACE		158 (12.5)	18 (11.4)	24 (15.2)	37 (23.4)	79 (50.0)	27.42	<0.001
	Myocardial infarction	34 (2.7)	2 (5.9)	5 (14.7)	10 (29.4)	17 (50.0)	9.25	0.025
	Ischemic stroke	43 (3.4)	6 (14.0)	8 (18.6)	13 (30.2)	16 (37.2)	5.48	0.138
	Death	113 (8.9)	15 (13.3)	14 (12.4)	25 (22.1)	59 (52.2)	22.71	<0.001

Notes: Data are expressed as the *n* (%). DM, diabetes mellitus; MACE, major adverse cardiovascular events; MALE, major adverse limb events.

**Table 3 jcm-11-02547-t003:** Univariate analysis of factors associated with hazard ratios of major adverse events.

Variables	MALE		MACE	
	Hazard Ratio (95% CI)	*p*	Hazard Ratio (95% CI)	*p*
Group				
A: non-DM/regular exercise	Reference		Reference	
B: non-DM/non-regular exercise	0.83 (0.61–1.13)	0.231	0.98 (0.53–1.80)	0.94
C: DM/regular exercise	1.20 (0.88–1.65)	0.247	2.07 (1.18–3.64)	0.01
D: DM/non-regular exercise	1.54 (1.18–2.03)	0.002	2.73 (1.63–4.56)	<0.001
Age, years	1.01 (1.00–1.02)	0.021	1.05 (1.03–1.07)	<0.001
Sex (Ref. male)				
Female	1.41 (1.13–1.76)	0.002	1.51 (1.06–2.17)	0.024
Smoking (Ref. never smoker)				
Current/former smoker	0.82 (0.68–0.99)	0.035	0.77 (0.56–1.06)	0.104
Critical limb ischemia	2.16 (1.79–2.61)	<0.001	2.51 (1.83–3.43)	<0.001
Hypertension	0.98 (0.80–1.19)	0.806	1.83 (1.25–2.69)	0.002
Dyslipidemia	0.87 (0.72–1.05)	0.147	0.87 (0.64–1.20)	0.389
Chronic kidney disease	1.84 (1.51–2.27)	<0.001	4.23 (3.09–5.79)	<0.001
Coronary artery disease	0.99 (0.82–1.20)	0.928	1.63 (1.19–2.23)	0.002
Congestive heart failure	1.70 (1.28–2.27)	<0.001	2.72 (1.79–4.11)	<0.001
Ischemic stroke	0.88 (0.63–1.23)	0.452	0.88 (0.50–1.55)	0.650
Duration of DM	1.05 (1.03–1.08)	<0.001	1.15 (1.10–1.19)	<0.001

Notes: CI, confidence interval; DM, diabetes mellitus; MACE, major adverse cardiovascular events; MALE, major adverse limb events; Ref., reference.

**Table 4 jcm-11-02547-t004:** Multivariate analysis of factors associated with hazard ratios of major adverse events.

Variables	MALE		MACE	
	Hazard Ratio (95% CI)	*p*	Hazard Ratio (95% CI)	*p*
Group				
A: non-DM/regular exercise	Reference		Reference	
B: non-DM/non-regular exercise	0.75 (0.55–1.02)	0.068	0.88 (0.48–1.64)	0.692
C: DM/regular exercise	0.92 (0.66–1.28)	0.600	1.25 (0.69–2.27)	0.460
D: DM/non-regular exercise	1.06 (0.77–1.43)	0.742	1.22 (0.68–2.19)	0.510
Age, years	1.08 (1.00–1.01)	0.201	1.05 (1.03–1.06)	<0.001
Sex (Ref. male)				
Female	1.31 (1.00–1.70)	0.050	1.31 (0.84–2.04)	0.228
Smoking (Ref. never smoker)				
Current/former smoker	1.11 (0.87–1.40)	0.411	1.26 (0.85–1.86)	0.254
Critical limb ischemia	1.85 (1.50–2.29)	<0.001	1.63 (1.14–2.33)	0.008
Hypertension	-		0.89 (0.58–1.35)	0.572
Chronic kidney disease	1.40 (1.11–1.77)	0.005	2.77 (1.91–4.01)	<0.001
Coronary artery disease	-		1.34 (0.94–1.90)	0.106
Congestive heart failure	1.36 (1.01–1.84)	0.049	1.31 (0.83–2.07)	0.241
Duration of DM	1.00 (0.96–1.03)	0.787	1.05 (1.00–1.10)	0.057

Notes: CI, confidence interval; DM, diabetes mellitus; MACE, major adverse cardiovascular events; MALE, major adverse limb events; Ref., reference.

## Data Availability

The data presented in this study are available on reasonable request from the corresponding author.

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
