# Peer review of "Major Adverse Events in Patients with Peripheral Artery Disease after Endovascular Revascularization: A Retrospective Study"

_jcm, 2022, doi:10.3390/jcm11092547_

Round 1
Reviewer 1 Report
Methods:
Of note, this is a prospective cohort study as baseline data were captured from endovascular revascularization hospitalization records, and patients were then followed until an event
Why ICD 10 code was not used to identify major amputation
Why only used ICD 10 codes to identify MI and stroke
How was the duration of DM calculated
Results:
The main analysis of DM/exercise and MALE/MACE was no longer significant in multivariable analyses. It would be interesting to treat exercise as a categorical variable and DM as a binary variable to see whether it's significantly associated with adverse events.
Conclusion:
The author emphasized this is the first study investigating DM/exercise and MALE/MACE, but did not summarize their findings (negative finding is also important to point out, and explain to readers why results were not significant after adjustment, several sentences of potential mechanism would also be helpful).
Author Response
- Methods: Of note, this is a prospective cohort study as baseline data were captured from endovascular revascularization hospitalization records, and patients were then followed until an event, Why ICD 10 code was not used to identify major amputation, Why only used ICD 10 codes to identify MI and stroke, How was the duration of DM calculated
Response: Thank you for your comment. This study conducted a retrospective cohort design for patients with peripheral artery disease after endovascular revascularization. EMR data from a single tertiary hospital used for analysis were recorded on the operation records using ICD-9-CM codes for major amputation, and the disease diagnosis was recorded using the ICD-10-CM codes in physicians’ progress notes. Therefore, based on the codes used in the hospital, ICD-9-CM was used for surgery/procedure names and ICD-10 CM for disease diagnosis.
In addition, the DM duration was calculated from the date of the first DM diagnosis to the index data (the date of the first endovascular revascularization).
We have added some text in the methods section (lines 102 ~ 104).
“In the hospital used for EMR analysis, the surgical/procedure code lists and disease diagnosis were recorded using ICD-9-CM and ICD-10-CM, respectively.”
- Results: The main analysis of DM/exercise and MALE/MACE was no longer significant in multivariable analyses. It would be interesting to treat exercise as a categorical variable and DM as a binary variable to see whether it's significantly associated with adverse events.
Response: Thank you for your comment. We analyzed again according to your recommendation. However, exercise was not statistically significant in major adverse events in multivariate analysis. Unfortunately, this result is similar to the initial results, therefore, we did not change the results.
- Conclusion: The author emphasized this is the first study investigating DM/exercise and MALE/MACE, but did not summarize their findings (negative finding is also important to point out, and explain to readers why results were not significant after adjustment, several sentences of potential mechanism would also be helpful).
Response: Thank you for your comment. We have added some text in the conclusion section (lines 241 ~ 246).
“In the present study, group differences were not statistically significant in MALE and MACE incidence after endovascular revascularization. Target lesion status (location, length of lesions, and degree of calcification) or clinical ischemia before the endovascular revascularization affects the vessel patency [2], associated with exercise maintenance and clinical prognosis. Thus, the CLI and CKD associated with the vascular condition before the procedure may be at high risk for the incidence of MALE and MACE.”

Reviewer 2 Report
Dear authors,
The study is a retrospetive study about the association between PAD and adverse events after endovaskular revascularization. The catagorization oft he groups based only on diabetes mellitus and regular exercise. The analysis of these groups were good.
But he study design is too weak for peripheral artery disease (PAD), because where is no information about the type of PAD:
There is no classification of the PAD e.g. according to Rutherford classification.
Clinical symptoms such as intermittent claudication, critical limb ischemia with rest pain and tissue loss are important for an outcome. In Table 1 were reported about „abnormal skin colour“ in „Clinical manifestation“. Please define it clearly. Do you mean tissue loss of the foot or toe?
Information about the target vessel for endovascular revascularization and information about the diseases segmet e.g. iliac artery, superficial artery or arteries below the knee were not reported. Which vessel and why was treated?
Information about the run-off vessels below the knee was missing, too.
Technical success and outcome depend on the lengh of the lesion. Please report about the type of lesion.
Regular exercise which is used to categorize the groups is not defined. Please define it in objective parameters. What do you use for the diagnosis (ultrasound, doppler, CTA, MRA...)?
Abstract: Please change the design of the abstract in "Objective, Patients and Methods,
Results, Conclusion",
Author Response
- The study is a retrospetive study about the association between PAD and adverse events after endovaskular revascularization. The catagorization of the groups based only on diabetes mellitus and regular exercise. The analysis of these groups were good. But he study design is too weak for peripheral artery disease (PAD), because where is no information about the type of PAD: There is no classification of the PAD e.g. according to Rutherford classification.
Response: Thank you for your comment. This study conducted a retrospective EMR review. EMR data from a single tertiary hospital used for analysis did not find Rutherford or Fontaine classification of all patients with PAD. Therefore, we extracted relevant information from EMR and classified clinical manifestations into asymptomatic, intermittent claudication, critical limb ischemia, and abnormal skin color (Line 86 ~ 89).
We have added some sentences in the discussion section (lines 294 ~ 296).
“In addition, in this study, due to the limitations of the relevant information stored in EMR, it was not reported as Rutherford or Fontaine classification, which classifies symptoms in PAD patients, and it was reported as clinical manifestation instead.”
- Clinical symptoms such as intermittent claudication, critical limb ischemia with rest pain and tissue loss are important for an outcome. In Table 1 were reported about „abnormal skin colour“ in „Clinical manifestation“. Please define it clearly. Do you mean tissue loss of the foot or toe?
Response: Thank you for your correction. “Abnormal skin color” means a color change of the toes without symptoms.
We have added some text in the methods section (lines 89 ~ 90).
“Abnormal skin color recorded in physicians’ progress notes means a color change of the toes without symptoms.”
Information about the target vessel for endovascular revascularization and information about the diseases segmet e.g. iliac artery, superficial artery or arteries below the knee were not reported. Which vessel and why was treated? Information about the run-off vessels below the knee was missing, too. Technical success and outcome depend on the lengh of the lesion. Please report about the type of lesion.
Response: Thank you for your suggestion. As you suggested, we have added some text and information in the results section (Table 1 and lines 142 ~ 146).
“There was a significant group difference in multilevel disease (p = 0.002). The aortoiliac lesions were more frequently in Group A, while infrapopliteal lesions were more frequently in Group D. The femoropopliteal lesions did not differ between the groups (p = 0.341). Balloon angiography was performed in approximately 97%, while stent insertion was performed in 56.6%.”
In addition, we conducted a Cox proportional hazards regression adding these variables (run-off vessels). However, the results were similar to the initial results. So, we did not change the results.
- Regular exercise which is used to categorize the groups is not defined. Please define it in objective parameters. What do you use for the diagnosis (ultrasound, doppler, CTA, MRA...)?
Response: Thank you for your concerns. In this study, regular exercise data did not use objective indicators (lines 284 ~ 286), and self-reported questions (lines 84 ~ 86) about whether the patient exercised regularly from the initial nursing evaluation record were extracted and used.
The included patients in this study satisfied both PAD diagnosis and endovascular revascularization as described in the methods sections (lines 68 ~ 70). Therefore, all patients were diagnosed and treated through percutaneous transluminal angioplasty (PTA).
- Abstract: Please change the design of the abstract in "Objective, Patients and Methods, Results, Conclusion",
Response: Thank you for your correction. We have changed the abstract to the structured format, as you suggested.

Round 2
Reviewer 1 Report
no further comments